# AstroCLIP: Cross-Modal Pre-Training for Astronomical Foundation Models

**Francois Lanusse**[*1,2]     **Liam Parker**[1]     **Siavash Golkar**[1]     **Miles Cranmer**[3]

**Alberto Bietti**[1]     **Michael Eickenberg**[1]     **Geraud Krawezik**[1]     **Michael McCabe**[1,4]

**Ruben Ohana**[1]     **Mariel Pettee**[1,5]     **Bruno Régaldo-Saint Blancard**[1]

**Tiberiu Tesileanu**[1]     **Kyunghyun Cho**[6,7,8]     **Shirley Ho**[1,6,9]

The Polymathic AI Collaboration

[1] Flatiron Institute     [2] Université Paris-Saclay, Université Paris Cité, CEA, CNRS, AIM

[3] University of Cambridge     [4] University of Colorado Boulder

[5] Lawrence Berkeley National Laboratory     [6] New York University

[7] Prescient Design, Genentech     [8] CIFAR Fellow     [9] Princeton University

## Abstract

We present AstroCLiP, a strategy to facilitate the construction of astronomical foundation models that bridge the gap between diverse observational modalities. In particular, we demonstrate that a cross-modal contrastive learning approach between images and spectra of galaxies yields highly informative embeddings of both modalities. We apply our method to multi-band images and spectrograms from the Dark Energy Spectroscopic Instrument (DESI), and show that: (1) these embeddings are well-aligned between modalities and can be used for accurate cross-modal searches, and (2) these embeddings encode valuable physical information about the galaxies - in particular redshift and stellar mass - that can be used to achieve competitive zero- and few- shot predictions without further finetuning. Additionally, we develop the first transformer-based model and pretraining approach for galaxy spectra. [2]

---

[*]Contact: flanusse@flatironinstitute.org

[2]Code: https://github.com/PolymathicAI/AstroCLIP

Submitted to NeurIPS 2023 AI for Science Workshop.

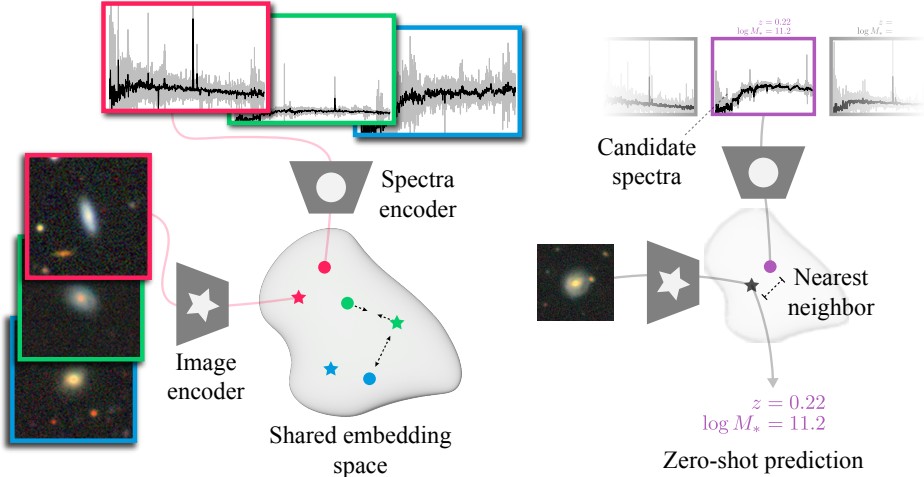

Figure 1: Illustration of the *AstroCLIP* cross-modal training strategy. We embed the images and optical spectra of galaxies into a shared embedding space, and use cross-modal contrastive learning to align these embeddings around shared semantics. Once trained, these embeddings allow us to connect and compare cross-modal representations. Moreoever, they possess physically meaningful high-level information which can then be used for a variety of downstream tasks.

# 1  Introduction

Astronomical datasets continue to enjoy a rapid expansion in size and complexity. However, the objects in these datasets often lack high-quality labels or representations, which makes the detailed analysis of such large volumes of data increasingly challenging for researchers. While crowd-sourced campaigns [e.g. Willett et al., 2013] have been used on the scales of tens of thousands of galaxies, they fall short by several orders of magnitude of the scale of modern surveys, which encompass tens of millions of objects for the ongoing Dark Energy Spectroscopic Instrument (DESI) [Dey et al., 2019] and several billions of objects for the upcoming Vera C. Rubin Legacy of Surveys of Space and Time (LSST) [Ivezić et. al., 2019].

As such, a wide array of computational approaches have been developed to help process the data from these surveys [Ivezić et al., 2020]. In recent years, a growing subset of these approaches have involved using advances in machine learning (ML) across a variety of classification or regression tasks (see Huertas-Company and Lanusse [2023] for a recent review). However, most of the proposed ML-backed models have relied on a supervised training regime and thus remain intrinsically limited by the quality and quantity of labelled training samples available.

To overcome this reliance on labeled data, researchers have developed a variety of approaches that do not rely on data labels for training. For example, unsupervised strategies with auto-encoders [e.g. Portillo et al., 2020] have been used to embed galaxy spectra into low-dimensional representations, and self-supervised contrastive learning - inspired by recent advances in computer vision [He et al., 2020, Chen et al., 2020a] - has been employed to extract semantically meaningful representations from galaxy images [Huertas-Company et al., 2023]. These embedded representations can then be used both in their own right - for similarity searches, outlier removal, etc. - and as "foundations" for downstream tasks [Hayat et al., 2020, Stein et al., 2021b].

To date, these approaches have been limited to embedding objects from a single modality, which typically relies on creating artificial augmented views of the same data. However, in an astrophysical context, there exist a number of complementary observations of the same physical objects (galaxies); for example, images obtained in different filters (i.e. at different optical wavelengths), or even different kinds of observations entirely such as optical spectra[3]. These different observational modalities can be thought of as different views of the same objects and as such, a universal embedding of these

---

[3]Optical spectra correspond to the flux of a galaxy as a function of the wavelength of its light.

objects should be able to simultaneously embed cross-modal representations of the same object into a shared embedding space.

**Contributions.** To that end, we introduce *AstroCLIP*, a cross-modal contrastive learning approach applied to astronomical datasets, inspired by the Contrastive Language-Image Pretraining model [Radford et al., 2021]. In particular, our approach takes information about galaxies from two separate modalities - images and optical spectra - and embeds and aligns both into a shared embedding space, which can be used for both cross-modal searches and as a foundation for downstream tasks, as illustrated in Fig. 1. Our specific contributions are:

- We develop AstroCLIP, an approach that embeds both the optical spectra and the images of galaxies into a shared embedding space, and aligns spectra and images of the same galaxy through self-supervised contrastive learning.
- We develop the first transformer-based model for galaxy spectra, along with an effective pre-training strategy for this model.
- We demonstrate that our cross-modal embeddings are well-aligned and can be used for accurate cross-modal searches.
- We demonstrate that our embeddings encode valuable high-level information that can be used for downstream tasks, as even simple techniques like k-nearest neighbor prediction can predict physical properties of galaxies like mass and redshift[4] from the learned embeddings.

## 2 Related work

**Contrastive Learning** In recent years, contrastive learning has emerged as an effective paradigm for learning meaningful representations of data in a self-supervised context. By bringing similar data closer and pushing dissimilar data apart in the embedding space, the model learns robust representations of the underlying data. The success has been exemplified both in single-modal tasks, like in computer vision [He et al., 2020, Chen et al., 2020b], as well as in cross-modal domains, where it has been used to connect language and image representations [Radford et al., 2021].

**Foundation Models for Astronomical Images** One of the earliest works in this direction is the application of large-scale, self-supervised contrastive learning to galaxy images using a MoCo framework [Hayat et al., 2020, Stein et al., 2021b]. This framwork learns embeddings which can be used to predict galaxy properties (such as redshift in Hayat et al. [2021]) as well as to perform similarity searches, including the identification of rare, scientifically interesting events like strong gravitational lenses [Stein et al., 2021a]. Another prominent example in this field is the application of a similar BYOL self-supervised training strategy [Grill et al., 2020] for pretraining networks than can further be easily fine-tuned, even in the low data regime, for the task of classifying galaxies according to their morphologies [Walmsley et al., 2022, Slijepcevic et al., 2023]. For a more detailed overview, we direct the reader to the recent review of contrastive learning in astrophysics from Huertas-Company et al. [2023].

**Representation Learning for Galaxy Spectra** Traditionally, techniques like Principal Component Analysis (PCA) have proven widely successful for extracting information from galaxy spectra. However, a new line of inquiry using unsupervised machine learning techniques has recently emerged. For example, Portillo et al. [2020] use a variational auto-encoder (VAE) to reduce the dimensionality of galaxy spectra to a small latent space, and demonstrate that the learned embeddings of the spectra can be used for downstream tasks like outlier detection, interpolation, and galaxy class classification. Teimoorinia et al. [2022] improve upon the existing VAE by introducing convolutional elements into the AutoEncoder to extract correlated features from the spectra. Melchior et al. [2023] add an attentive convolutional encoder and include elements of physical modeling of observational factors into the AutoEncoder; their embeddings are then similarly useful for downstream tasks like anomaly detection [Liang et al., 2023b,a]. Presently, only a couple of attempts have been made so far at connecting images and spectra, but only from the point of view of attempting to generate spectra conditioned on images [Wu and Peek, 2020, Doorenbos et al., 2022].

---

[4]The particular redshift corresponds to the distance of the galaxy from the observer. Galaxies that are further away will see their spectrum shifted further red.

# 3 Methodology

## 3.1 Training Objective

At the core of our work is the idea that different observational modalities can be thought of as filtered views of the same underlying physical objects and therefore intrinsically possess a shared physical latent space. Thus, we aim to construct embeddings of both modalities that maximize the mutual information about the underlying object, and then to use that mutual information to align representations from different modalities around shared semantics.

Formally, let $\mathbf{x} \in \mathbb{R}^N$ and $\mathbf{y} \in \mathbb{R}^M$ be two examples from two different modalities. We aim to construct a pair of models $f_\theta : \mathbb{R}^N \to \mathbb{R}^d$ and $g_\theta : \mathbb{R}^M \to \mathbb{R}^d$ that compress these two examples to a shared $d$-dimensional space such that the mutual information between these representations, $I(f_\theta(\mathbf{x}), g_\theta(\mathbf{y}))$, is maximized. To that end, we employ contrastive training under an InfoNCE loss [van den Oord et al., 2018, Gutmann and Hyvärinen, 2010], where we consider that both modalities represent noisy views of the same underlying physical object. This loss is given by

$$\mathcal{L}(\mathbf{x}, \mathbf{y}, \tau) = -\sum_i \log \frac{\exp(f_\theta(\mathbf{x}_i)^t g_\theta(\mathbf{y}_i)/\tau)}{\exp(f_\theta(\mathbf{x}_i)^t g_\theta(\mathbf{y}_i)/\tau) + \sum_{i \neq j} \exp(f_\theta(\mathbf{x}_i)^t g_\theta(\mathbf{y}_j)/\tau)} \ , \tag{1}$$

where $\tau$ represents a smoothing parameter (sometimes referred to as temperature) and $j$ represent the indices of negative examples, not associated with the object $i$. We consider the spectrum and the image of the same object a pair, as exemplified by objects highlighted by the same color on Fig. 1, and all other combinations of spectra and images from different galaxies to be negative samples.

Ultimately, training under this objective should extract embeddings from both modalities that contain the shared physical information between the galaxy images and spectra. Additionally, as shown by a variety of examples in computer vision, the learned embeddings typically exhibit highly structured information about the underlying object, going beyond the strict intersection of information between modalities. This is an emerging property we hope to witness in our model as well.

## 3.2 Implementation

As our representation pairs are different modalities, we deploy a pair of models to perform the embeddings. We start with a pretrained image embedder and a pretrained spectrum embedder which embed the galaxy image/spectrum into a shared embedding space $Z \in \mathbb{R}^{128}$. We then fine-tune both models using the contrastive InfoNCE loss to align the embedded spectra and image of the same galaxy around shared semantics. The details of both pretrained models and the alignment strategy are given below.

**Pretrained spectrum embedder:** As our architecture for the spectrum embedder, we adopt a transformer model structured similar to GPT-2 [Radford et al., 2019][5]. To format the spectra appropriately for the transformer, we first reshape their T dimensional native representation (where T $\approx$ 7,000) to a sequence of shape $(T \bmod 10) \times 20$, where each element of this new sequence is a contiguous 20-element segment of the original sequence, and adjacent elements have an overlap of size 10. Since the dataset includes samples of highly different overall amplitudes, in order to make it easier for the network to process all samples, we Z-score each individual sample. We include the mean ($\mu$) and standard deviation ($\sigma$) information by appending it to the sequence as follows: Using a sequence of length $x = (1 + T \bmod 10) \times 22$, we embed $\mu$ and $\sigma$ in the first element ($x_{0,0} = \mu, x_{0,1} = \sigma$) and let $x_{1:,2:}$ be equal to the Z-scored $(T \bmod 10) \times 20$ sequence described above. After this reformatting step, we use a transformer with embedding dimension 768, 6 transformer block layers with 6 heads, totalling 43.2M parameters.

We pretrain this transformer only on spectra first, using a self-supervised learning paradigm. We randomly replace 6 contiguous segments of length 30 (equivalent to length 600 in the original spectra

---

[5]In particular, we use absolute positional embeddings and a pre-attention layer norm. However, we deviate from GPT-2 in that we initialize all the weights of the transformer blocks with a normal distribution with standard deviation given by $(2 \times \text{fan-in} \times \text{num-layers})^{-1/2}$. The dependence of the standard deviation on the number of transformer blocks is to counteract the effect of having a series of residual connections.

representation) with zeros and train the model to minimize the Mean Square Error loss between the predictions and the ground truth on the replaced segments of the sequence. For details on the performance of this mask-filling model see Supplementary Materials Appendix C.

Once this mask-filling model has been trained, we freeze its weights and use a single cross-attention block (cross-attention layer with 4 heads and embedding dim 128 followed by an MLP) to extract a short embedding vector. We use the output of the final transformer block of the mask-filling model as the key and value and use a learnable sequence of size $1 \times 128$ as the query vector. The output of this procedure is a single vector of length 128. The weights of this cross-attention block and the 128 parameters of the query vector amount to 362k total parameters which are then trained via the contrastive training training procedure described below.

**Pretrained image embedder:** We use the pretrained galaxy image encoder from Stein et al. [2021a]. This model is based on MoCo V2 [Chen et al., 2020b], and uses a ResNet50 backbone as the encoder. The model is pretrained in a self-supervised regime, using augmented pairs of galaxies that have undergone a variety of augmentations, including galactic extinction, rotation, size scaling, point-spread function blur, jittering and cropping, and Gaussian noise. Pretraining is performed on a curated subset of 3.5 million galaxies sampled uniformly by z-band magnitude from the the DESI Legacy Survey. For more details on the model, we refer the reader to Stein et al. [2021a]. This model has 28M parameters in total, during our contrastive training phase, we keep the convolutional part of the model frozen and only finetune the dense final layers which amount to 4.5M parameters.

**Contrastive Training:** The pretrained models are frozen and fed into our unified AstroCLIP model. We unfreeze the fully connected layers of the pretrained image encoder and attach an additional trainable single-layer, four-head transformer to the pretrained spectrum encoder. Both models are then trained such that the embedded representations $\mathbf{z}$ for each galaxy image/spectrum are aligned. This training is performed using the InfoNCE objective of Equation 1, where embedded representations are considered pairs if they pertain to the same galaxy, and are considered negative examples if they pertain to different galaxies. We set the queue length to $K = 512$ image-spectrum pairs, and perform basic data augmentation with random vertical and horizontal flips, random rotationson the images. We train our models for 15,000 iterations, which takes roughly 5 hours on a single h100 gpu. Finally, similarly to findings of similar works [Girdhar et al., 2023] we find better performance by fixing the value of the temperature parameter $\tau$ as opposed to letting it free.

### 3.3 Data

For this work, we use the DESI Legacy Survey [6] Data Relase 9 imaging data [Dey et al., 2019] as prepared by Stein et al. [2021b]. This corresponds to an initial set of 41 million $(g, r, z)$ $152 \times 152$ images, which we center crop to $96 \times 96$. In complement, we cross-match galaxy spectra from the DESI Early Data Release [Collaboration et al., 2023], which yields a total subset of 197,976 pairs of images and spectra. Finally, for the experiments involving the extraction of physical information from these embeddings, we further cross-match this sample with physical properties for these galaxies reported in the PRObabilistic Value-Added Bright Galaxy Survey (PROVABGS) Catalog from Hahn et al. [2023].

## 4 Results

### 4.1 Example retrieval by cosine similarity

To visualize our embedding scheme's ability to align representations of galaxies, we query galaxies and find the nearest neighbors in our embedding space using a cosine similarity search. For example, the cross-modal similarity $S_C(\mathbf{z}_i^{sp}, \mathbf{z}_j^{im})$ between a query spectrum $\mathbf{z}_i^{sp} = g_\theta(\mathbf{x}_i^{sp})$ and an image $\mathbf{z}_j^{im} = f_\theta(\mathbf{x}_j^{im})$ is given computed as:

$$S_C(\mathbf{z}_i^{sp}, \mathbf{z}_j^{im}) = (\mathbf{z}_i^{sp} \cdot \mathbf{z}_j^{im}) / \parallel \mathbf{z}_i^{sp} \parallel \parallel \mathbf{z}_j^{im} \parallel \tag{2}$$

---

[6]https://www.legacysurvey.org/

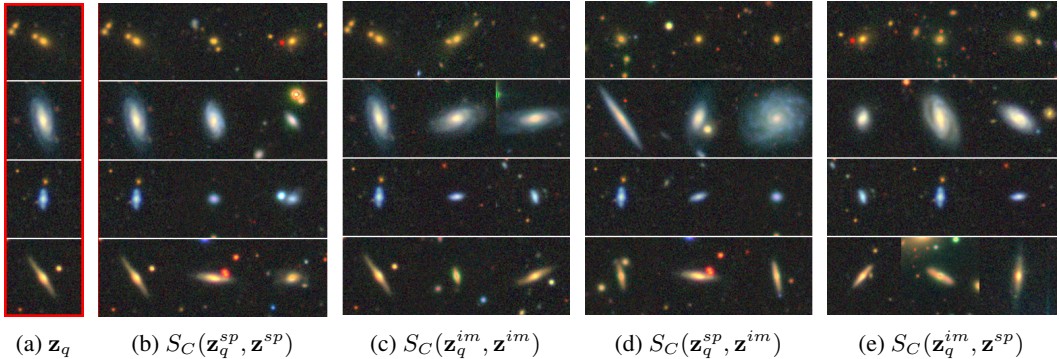

(a) $\mathbf{z}_q$    (b) $S_C(\mathbf{z}_q^{sp}, \mathbf{z}^{sp})$    (c) $S_C(\mathbf{z}_q^{im}, \mathbf{z}^{im})$    (d) $S_C(\mathbf{z}_q^{sp}, \mathbf{z}^{im})$    (e) $S_C(\mathbf{z}_q^{im}, \mathbf{z}^{sp})$

Figure 2: Example retrieval from both in-modality and cross-modality examples. For the experiments involving spectra, we only show here the paired image associated to the spectrum.

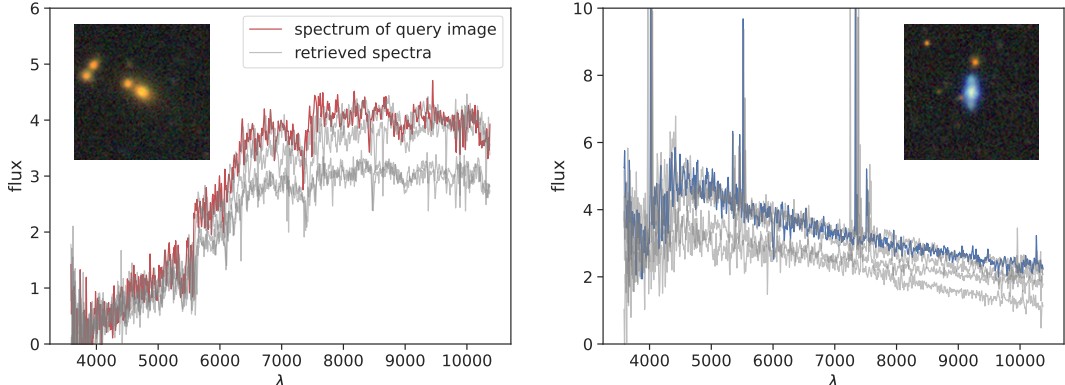

Figure 3: Retrieved spectra for a randomly chosen query image for two different query galaxies. The spectra are found using a cosine similarity search between the AstroCLIP embedding of the query image and those of the spectra in our dataset.

This is performed for both **in-modality similarity search**, where we determine the neighbors according to the similarity between embeddings derived from the same modality (i.e. image-image or spectrum-spectrum), and **cross-modality similarity search**, where we consider the similarity between embeddings derived from different modalities (i.e. image-spectrum or spectrum-image). These are presented for all four possible embedding pairs in Fig. 2 for a set of four randomly selected query examples. Ultimately, these examples demonstrate that the model is able to represent the same types of objects similarly, regardless of the original modality in which that object is represented. We note that, by construction, the closest match for an in-modal similarity search is indeed the object itself.

Additionally, we further illustrate the result of in-modal retrieval and cross-modal retrieval in Fig. 6 and Fig. 3 respectively, where we present the retrieved spectra for a randomly chosen query spectrum/image for two different query galaxies. These results demonstrate a strong correlation between the semantic content of the image, such as the red quiescent galaxy or a blue star forming galaxy, and the shape of the recovered spectra.

### 4.2 Zero-shot regression of physical properties

To reach more quantitative statements about the performance of AstroCLIP pretraining, we consider our ability to perform zero-shot prediction on a variety of downstream tasks from the embedded galaxy samples. In particular, we use simple k-Nearest Neighbour (k-NN) regression of our embedded images and spectra to infer the particular redshift and the stellar mass of our galaxies. Specifically, k-NN regression is performed on the autocorrelated AstroCLIP image and spectrum embeddings, as well as on the correlated image-spectrum embeddings.

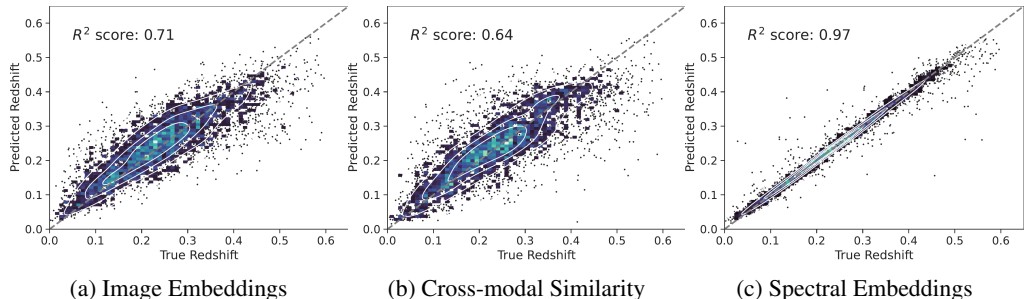

(a) Image Embeddings      (b) Cross-modal Similarity      (c) Spectral Embeddings

Figure 4: Redshift regression by k-NN for both in-modality and cross-modality similarity. In (b) the query domain is images.

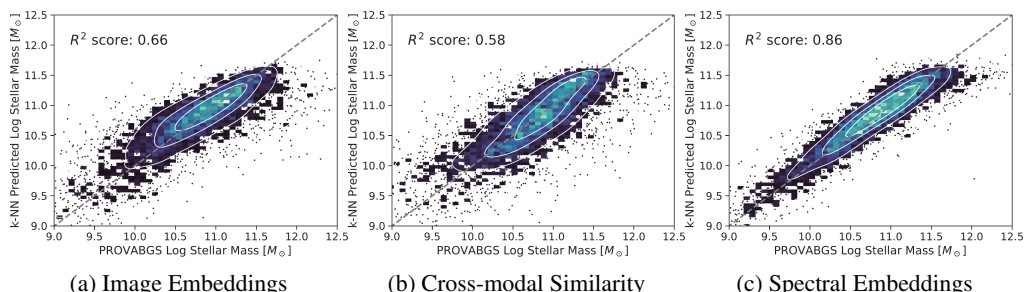

(a) Image Embeddings      (b) Cross-modal Similarity      (c) Spectral Embeddings

Figure 5: Stellar mass regression by k-NN for both in-modality and cross-modality similarity. In (b) the query domain is images.

We present in Fig. 4 and Fig. 5 a comparison of the performance of k-NN regression from the AstroCLIP embeddings for either in-modality or cross-modality similarity. We note a variety of important observations. For one, neighbors in our embedded space indeed share similar physical properties as demonstrated by the ability of our k-NN regressor to make accurate predictions. This indicates that our model is able to organize our galaxy samples according to high-level, physically meaningful features. Additionally, in-modality similarity appears to outperform cross-modality similarity as an input for the k-NN regression, indicating that, although our our contrastive training aims to connect embeddings between modalities, it has the emergent property of helping to structure the embeddings space within respective modalities. This is particularly evident for the redshift prediction by similarity between spectra which is near perfect, even though redshift is not an information perfectly contained in images. This means that redshift has naturally emerged as a fundamental property which helps the spectral encoder to structure its embedding space.

We present the numerical results of our k-NN regression on both the AstroCLIP image embeddings and spectrum embeddings in Tab. 1. We compare our results to the predictions of an MLP from 3-band photometry alone, i.e. a model which is only sensitive to the overall flux of the galaxy in each color band and ignores any aspects of morphology. Additionally, we compare our results to the out-of-the-box image embeddings of the ResNet-50 pretrained model from [Stein et al., 2021a], which was pretrained in a single-modal contrastive setting on augmentations of the images alone.

Ultimately, we demonstrate that our image embeddings are roughly as informative as simple photometry for the tasks at hand. Conversely, these embeddings far outperform the self-supervised pretraining approach from Stein et al. [2021b] without the need for any further finetuning. Interestingly, we note that our zero-shot approach (using the k-NN) outperforms our few-shot approach with the MLP; this occurs due to the fact that the k-NN results are tighter but slightly biased, a phenomena which is not fully reflected and penalized in the $R^2$ metric. Finally, we note that our spectrum embeddings outperform all other embeddings, and reach close to perfect accuracy on the redshift regression.

| Regression method | Redshift $R^2$ | Stellar Mass $R^2$ |
|---|---|---|
| $(r, g, z)$ Photometry + MLP | 0.69 | 0.56 |
| Stein et al. [2021b] Image Embedding + MLP | 0.39 | 0.45 |
| Image Embedding + k-NN (**ours**) | 0.71 | 0.66 |
| Spectrum Embedding + k-NN (**ours**) | 0.97 | 0.86 |
| Image Embedding + MLP (**ours**) | 0.63 | 0.57 |
| Spectrum Embedding + MLP (**ours**) | 0.99 | 0.86 |

Table 1: Performance of different regression methods, including our zero-shot learning k-NN prediction, on predicting two different physical properties of galaxies, their redshift (which corresponds to the distance from the observer) and their stellar mass. (Higher metrics are better)

## 5   Discussion

Our results demonstrate the potential for cross-modal contrastive pre-training to achieve high quality foundation models for astronomical data, which can be used for further downstream tasks even without fine-tuning. We contend that this is a key property to allow the community to build higher-level compositional models that can rely on off-the-shelf frozen embedding models, just as frozen CLIP embeddings have enabled a wide variety of downstream applications.

Reinforcing our optimism for this approach, our results also show that even if diverse modalities are not perfectly informative about each other, the contrastive learning task still allows the embedding of each modality to discover relevant physical patterns in the data. This is exemplified by the fact that our spectral embeddings exhibit an emergent ability to retain information about redshift that extends beyond the information captured in the images. This opens an interesting avenue for further research to build informative embeddings from a wider array of data modalities, even in the absense of strong connection between each of the individual modalities.

Finally, our work is one of the first to describe a transformer-based model for the modeling of galaxy spectra. We believe that the success of our experiment will pave the way for a broader adoption of transformer-based architectures for similar tasks, which have so far been overwhelmingly perfomed using 1D convolutional models. This transition may also facilitate the scaling of these models to the volume of data promised by DESI and LSST.

## Acknowledgments and Disclosure of Funding

We gratefully acknowledge the Flatiron Institute for its support. The computations in this work were run at facilities supported by the Scientific Computing Core at the Flatiron Institute, a division of the Simons Foundation. M.P. is supported by the Department of Energy, Office of Science under contract number DE-AC02-05CH11231.

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

## A  Additional results

Fig. 6 shows examples of retrieval using similarity of the spectrum encoding.

## B  Attention maps of spectra encoder

We look at the attention maps of the cross-attention layer of the spectrum encoder, described in Sec. 3.2. These plots can help interpret what information the model is looking at when building its represenation of the spectrum.

Fig. 7 shows a number of examples of these attention maps. We see that the different attention heads have specialized to look for different features. Head 1 seems to be looking at the two extremes of the spectrum which would make it sensitive to different spectral tilts. Head 3 seems to be sensitive to peaks around the 9k$\mathring{A}$ range. However, it is important to note that this cross-attention layer comes after the 6 layers of self-attention of the pre-trained model. At this stage of the network, information about different sections of the spectrum have likely diffused throughout the entire sequence and therefore the attention maps potentially access information from parts of the spectrum where the attention is zero.

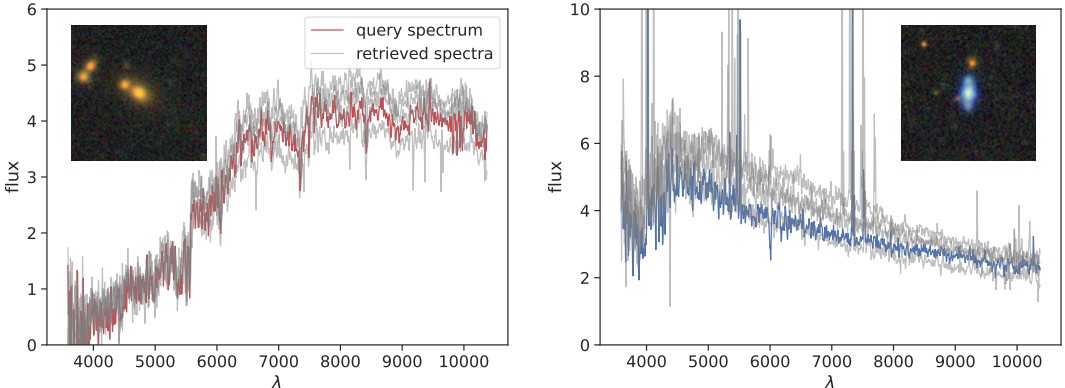

Figure 6: Retrieved spectra for a randomly chosen query spectrum for two different query galaxies. The spectra are found using a cosine similarity search between the AstroCLIP embedding of the query spectrum and those of the spectra in our dataset.

## C  Performance of the Mask-Filling spectrum encoder

The performance of the mask-filling model pretrained on the spectra can be seen in Fig. 8 to Fig. 12. In these figures, the shaded region denotes the area where the spectrum was zerod out when passed to the model. The various inserts show close-ups of the smoothed ground-truth (by taking averages of 20 bins) as well as the prediction of the model. We see that the model has learned to reproduce the prominent features of the spectru. For example, in both Fig. 8 and Fig. 12 a number of the masked regions have fallen on absorption and emmission lines. We see that the model can reproduce these features with high precision.

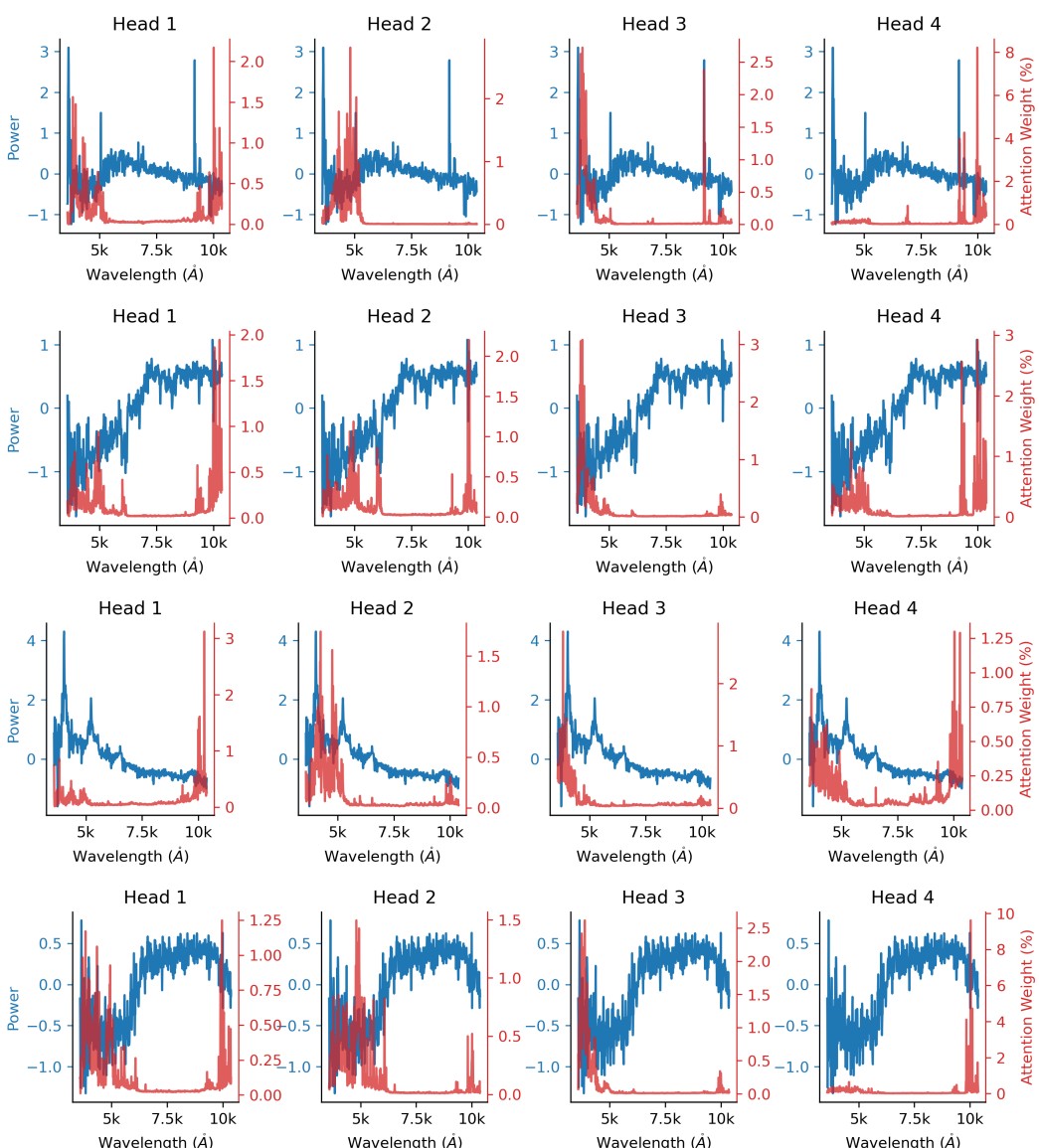

Figure 7: Examples of attention maps of the cross-attention layer of the spectrum encoder.

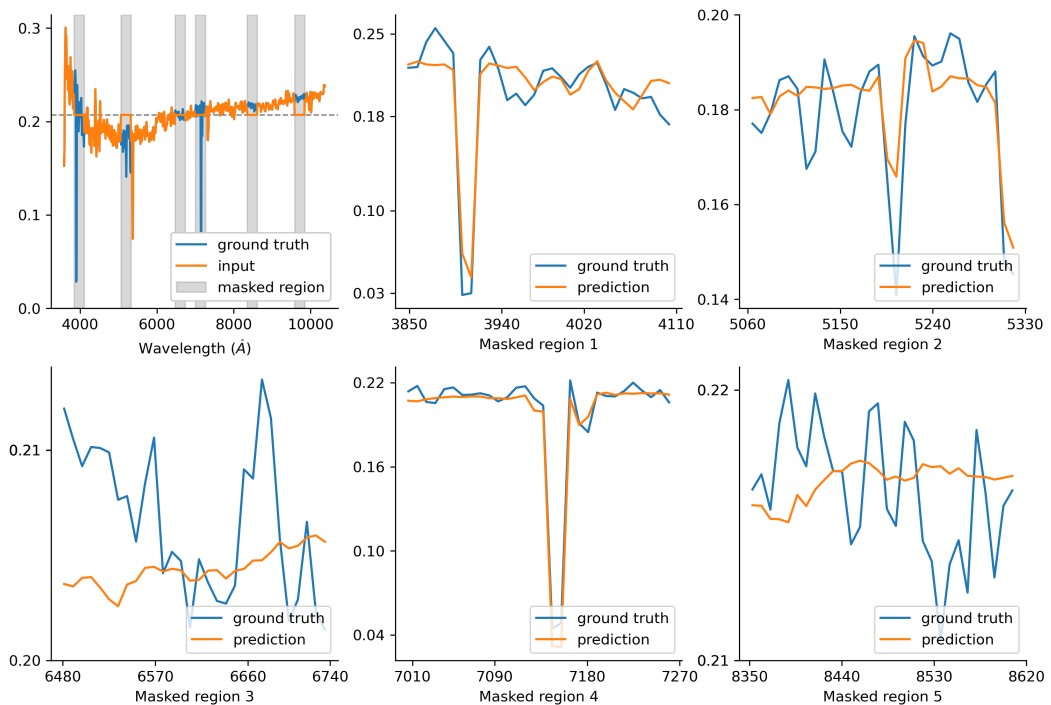

Figure 8: Example of the performance of the mask filling model.

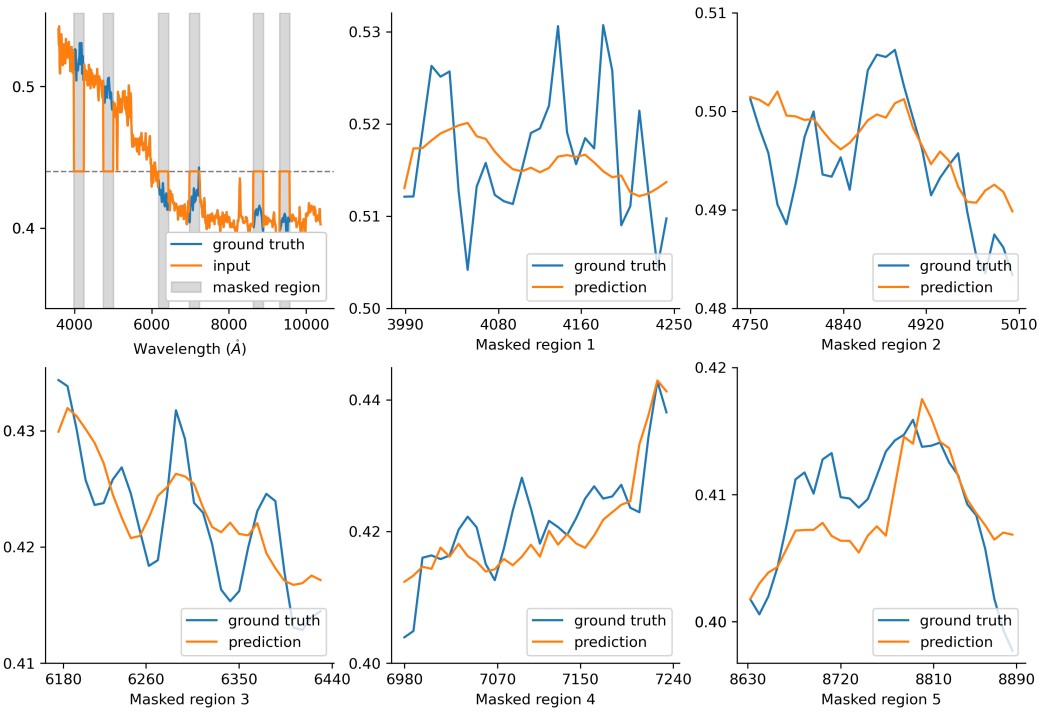

Figure 9: Example of the performance of the mask filling model.

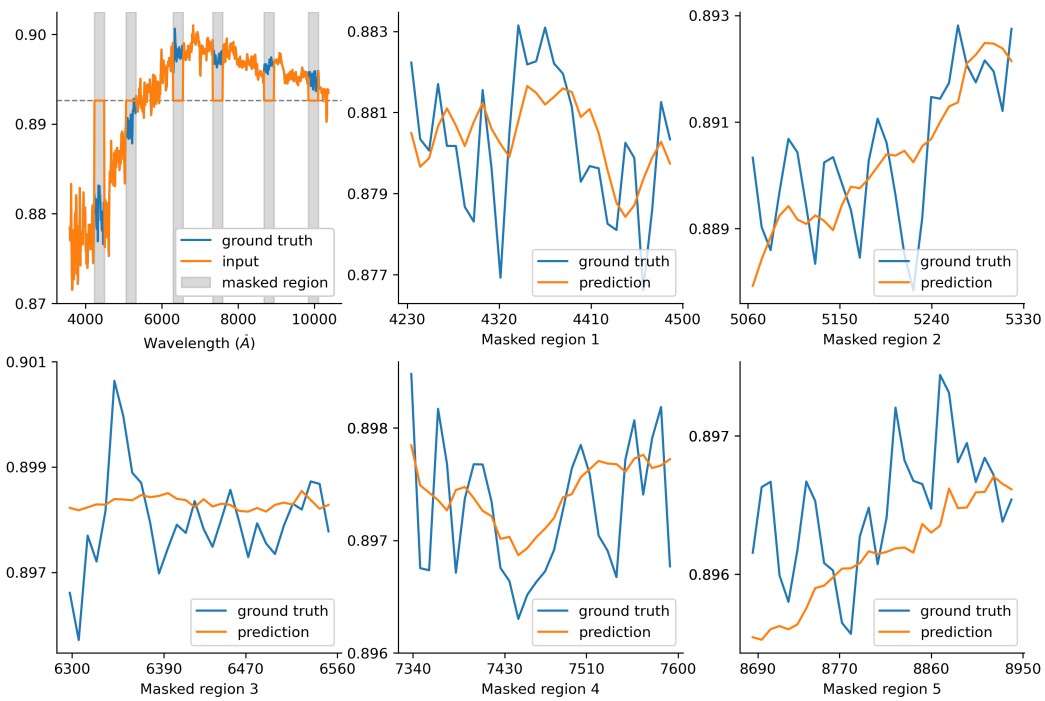

Figure 10: Example of the performance of the mask filling model.

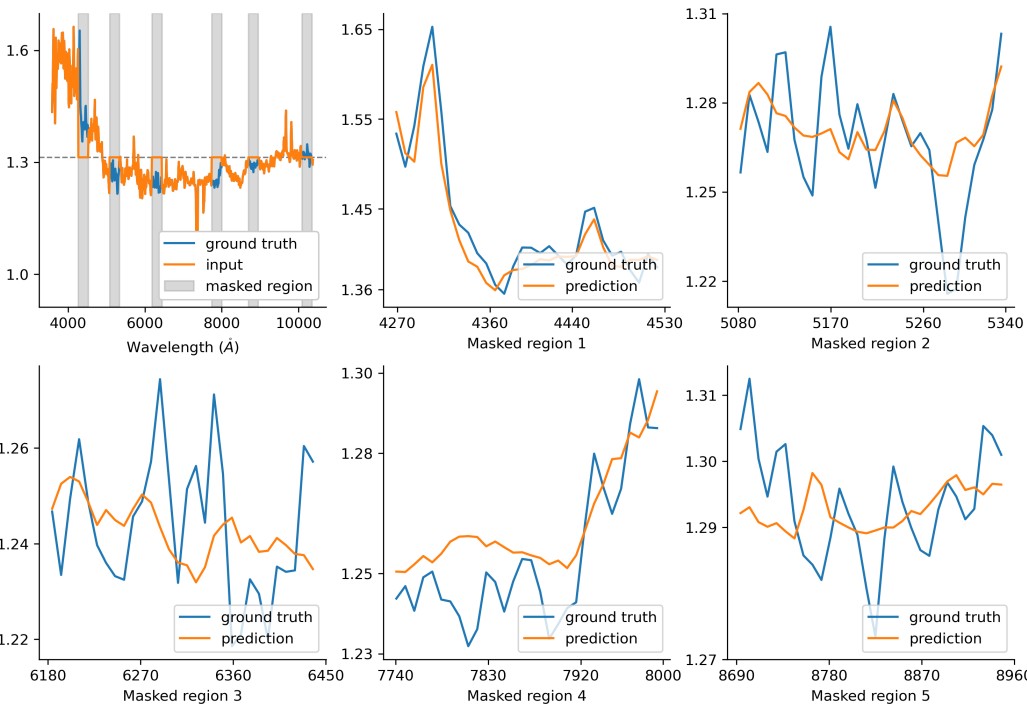

Figure 11: Example of the performance of the mask filling model.

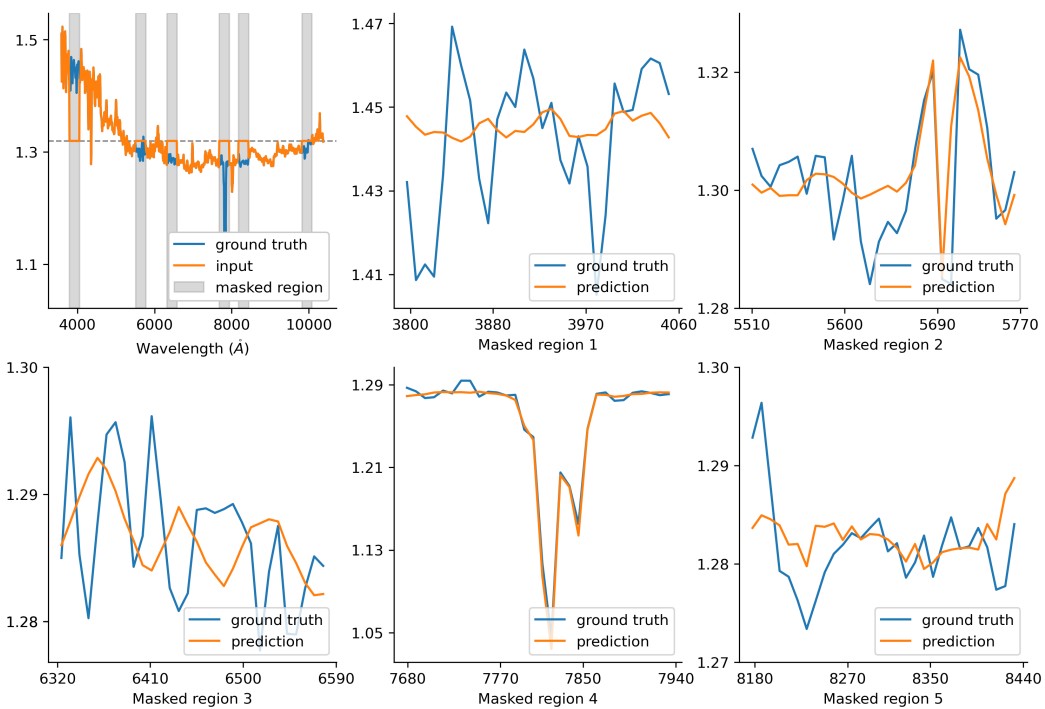

Figure 12: Example of the performance of the mask filling model.

