# OpenReview forum: "AstroCLIP: Cross-Modal Pre-Training for Astronomical Foundation Models"
_NeurIPS.cc/2023/Workshop/AI4Science — NeurIPS2023-AI4Science Poster_

### Official Review · Reviewer_dXyJ · 2023-10-15
**Good work that take advantage of data wealth in astrophysics**

**Rating:** 7
**Confidence:** 3

**Review:**

This a good paper that demonstrates that models pretrained on large astrophysics databases can outperform conventional ML approaches  for astrophysics ML tasks.

---

### Meta-Review · Area_Chair_Eavq · 2023-10-27

**Recommendation:** Accept (Poster)
**Confidence:** 3

**Metareview:**

The authors introduce a CLIP-inspired foundational model tailored for astronomy, trained on multi-band images and DESI spectrograms using InfoNCE loss. The study observes a synergistic effect with the inclusion of more modalities. This paper is likely to offer valuable insights to the research community.